# Complete chloroplast genomes of *Zingiber montanum* and *Zingiber zerumbet*: Genome structure, comparative and phylogenetic analyses

**Dong-Mei Li** *, Yuan-Jun Ye, Ye-Chun Xu, Jin-Mei Liu, Gen-Fa Zhu***

Guangdong Key Lab of Ornamental Plant Germplasm Innovation and Utilization, Environmental Horticulture Research Institute, Guangdong Academy of Agricultural Sciences, Guangzhou, China

* biology.li2008@163.com (DML); genfazhu@163.com (GFZ)

**Data Availability Statement:** We deposited the raw Illumina and PacBio reads into the NCBI. The Z. montanum chloroplast sequencing data have SRA numbers SRR8185396 and SRR8184511.

## Abstract

*Zingiber montanum* (*Z. montanum*) and *Zingiber zerumbet* (*Z. zerumbet*) are important medicinal and ornamental herbs in the genus *Zingiber* and family Zingiberaceae. Chloroplast-derived markers are useful for species identification and phylogenetic studies, but further development is warranted for these two *Zingiber* species. In this study, we report the complete chloroplast genomes of *Z. montanum* and *Z. zerumbet*, which had lengths of 164,464 bp and 163,589 bp, respectively. These genomes had typical quadripartite structures with a large single copy (LSC, 87,856–89,161 bp), a small single copy (SSC, 15,803–15,642 bp), and a pair of inverted repeats (IRa and IRb, 29,393–30,449 bp). We identified 111 unique genes in each chloroplast genome, including 79 protein-coding genes, 28 tRNAs and 4 rRNA genes. We analyzed the molecular structures, gene information, amino acid frequencies, codon usage patterns, RNA editing sites, simple sequence repeats (SSRs) and long repeats from the two chloroplast genomes. A comparison of the *Z. montanum* and *Z. zerumbet* chloroplast genomes detected 489 single-nucleotide polymorphisms (SNPs) and 172 insertions/deletions (indels). Thirteen highly divergent regions, including *ycf1*, *rps19*, *rps18-rpl20*, *accD-psaI*, *psaC-ndhE*, *psbA-trnK-UUU*, *trnfM-CAU-rps14*, *trnE-UUC-trnT-UGU*, *ccsA-ndhD*, *psbC-trnS-UGA*, *start-psbA*, *petA-psbJ*, and *rbcL-accD*, were identified and might be useful for future species identification and phylogeny in the genus *Zingiber*. Positive selection was observed for *ATP synthase* (*atpA* and *atpB*), *RNA polymerase* (*rpoA*), *small subunit ribosomal protein* (*rps3*) and other protein-coding genes (*accD*, *clpP*, *ycf1*, and *ycf2*) based on the Ka/Ks ratios. Additionally, chloroplast SNP-based phylogeny analyses found that *Zingiber* was a monophyletic sister branch to *Kaempferia* and that chloroplast SNPs could be used to identify *Zingiber* species. The genome resources in our study provide valuable information for the identification and phylogenetic analysis of the genus *Zingiber* and family Zingiberaceae.

The Z. zerumbet chloroplast sequencing data have SRA numbers SRR8185094 and SRR8184512. The final assembled chloroplast genomic sequences have been submitted to GenBank under accession numbers MK262727 and MK262726 for Z. montanum and Z. zerumbet, respectively. The bioproject number is PRJNA498576. We have also added the bioproject number in the manuscript. At this time, we declare that once this manuscript has published, all these relevant data were as soon as relieved from GenBank database.

**Funding:** This work was financially supported by Guangzhou Municipal Science and Technology Project (No.201607010101), National Natural Science Foundation of China (No.31501788), Guangdong Science and Technology Project (No.2015A020209078) and the special financial fund of Foshan–Guangdong Agricultural Science and technology demonstration city project in 2019.

**Competing interests:** The authors have declared that no competing interests exist.

## Introduction

*Zingiber* Boehm., belonging to the family Zingiberaceae, consists of between 100 and 150 species, all of which are widely distributed in southern and southeastern Asia, with particular concentrations in Thailand and southern China [1–4]. There are more than 40 *Zingiber* species in China, among which 13 are reported to have medicinal value [1, 2, 5]. In addition, most species have an assemblage of tightly clasped, overlapping bracts that often age to yellow, red, or chestnut brown and are often highly showy and long-lived, leading to the cultivation of a number of species for landscaping and cut-flower uses [2–4]. Both *Zingiber montanum* (J. König) A. Dietr and *Zingiber zerumbet* (Linnaeus) Rosc. ex Smith are useful medicinal and ornamental plants in this genus [2–5]. *Z. montanum* is endemic to the Guangdong, Guangxi, Hainan and Yunnan provinces of China [4]. Chemical compositions of the *Z. montanum* rhizome have antidiarrheal, antioxidant, antibacterial, antifungal, allelopathic and acetylcholinesterase inhibitory properties [3, 4, 6–8]. *Z. zerumbet*, commonly known as "shampoo ginger", is found across southern China (Guangdong, Guangxi, Hainan and Yunnan provinces), most of Southeast Asia, Myanmar, India, and Sri Lanka [1–4]. Zerumbone from the *Z. zerumbet* rhizome has been reported to suppress the phagocytic activity of human neutrophils [9], to prevent and treat tooth decay disease [10], to cure osteoarthritis of the knee [11], and to treat various immune-inflammatory related disorders [12].

*Zingiber* species have been known taxonomically, with many species based on both vegetative and floral characteristics [1–5]. However, a number of defining morphological features are often inconsistent and variable [1–4, 13]. Visually, *Zingiber* species are relatively similar to one another's vegetative parts in nonflowering seasons [1–4], making it highly difficult to morphologically distinguish among species in the nonflowering stage. Recently, several studies have also used molecular data to identify some *Zingiber* species [13, 14]. The results showed a weak resolution among six *Zingiber* species (*Zingiber corallinum*, *Zingiber wrayi*, *Zingiber sulphureum*, *Zingiber gramineum*, *Zingiber ellipticum* and *Zingiber species*) using nuclear internal transcribed spacer (*ITS*) and chloroplast *matK* regions [13]. Through amplified fragment length polymorphism (AFLP)-based DNA markers, the results have indicated that *Z. montanum* and *Z. zerumbet* are phylogenetically closer to each other than to *Zingiber officinale* [14]. These analyses have succeeded in clarifying the phylogenetic relationships and degrees of variation among *Zingiber* species, but in general have been limited in breadth of resolution. Therefore, a more accurate method of plant identification is essential for *Zingiber* species. The complete chloroplast genome contains more effective DNA markers, such as single-nucleotide polymorphisms (SNPs), insertion/deletions (indels) and hotspot variable regions, which can be used for accurate species identification. In recent years, more than 25 complete chloroplast genomes have been sequenced in the family Zingiberaceae [15–26]. However, to the best of our knowledge, the chloroplast genomes of *Z. montanum* and *Z. zerumbet* have not yet been elucidated. To date, only two *Zingiber* species' whole chloroplast genomes have been reported, namely, *Zingiber spectabile* (GenBank JX088661) and *Z. officinale* (NC_044775) [18], hindering the molecular plant identification of *Zingiber* species.

Chloroplasts are photosynthetic organelles that can transform light energy into chemical energy in green plants [27–29]. These organelles have their own chloroplast genomes that encode 110–130 genes with a size range of 120–180 kb and have a typical quadripartite structure consisting of a large single copy (LSC) region, a small single copy (SSC) region, and two copies of inverted repeats (IRs) [18–26]. Whole chloroplast genomes have been widely exploited to resolve plant phylogenies, origin problems and species identification [15–17, 22–26, 30].

In this study, we first sequenced and assembled the complete chloroplast genomes of *Z. montanum* and *Z. zerumbet* using combinations of Illumina and PacBio sequencing platforms, respectively. Second, we explored the molecular features of each genome and compared them with eight other members of the family Zingiberaceae. Third, we analyzed the codon usage, RNA editing, SNPs and indels in the chloroplast genome sequences of *Z. montanum* and *Z. zerumbet*. Fourth, we detected simple sequence repeats (SSRs), long repeats, highly divergent hotspot regions and phylogenetic relationships of *Z. montanum* and *Z. zerumbet* and compared them with two reported *Zingiber* species (*Z. officinale* and *Z. spectabile*). Our findings are expected to be useful for species identification and phylogenetic studies in the genus *Zingiber* and family Zingiberaceae.

## Materials and methods

### Ethical statement

No specific permits were required for the collection of specimens for this study. This research was carried out in compliance with the relevant laws of China.

### Plant material, chloroplast DNA extraction and sequencing

Fresh leaves were collected from *Z. zerumbet* and *Z. montanum* plants from the resource garden of the environmental horticulture research institute (23˚ 23' N, 113˚ 26' E), Guangdong Academy of Agricultural Sciences, Guangzhou, China. Total chloroplast DNA was extracted from these leaves using the improved sucrose gradient centrifugation method [31]. The quality and quantity of extracted chloroplast DNA were estimated using an ND-2000 spectrometer (Wilmington, DE, USA) and 1% agarose gel electrophoresis, respectively. Chloroplast DNA samples of good integrity with both optical density (OD) 260/280 and OD 260/230 ratios greater than 1.8 were used for sequencing.

Two libraries with insert sizes of 300 bp and 10 kb were constructed after DNA purification for each sample. Then, the samples were sequenced on an Illumina HiSeq X Ten instrument (Biozeron, Shanghai, China) and a PacBio Sequel platform (Biozeron, Shanghai, China), respectively. The qualities of Illumina raw reads and PacBio raw reads were determined using FastQC. After filtering the raw data, 43.4 M and 73.9 M clean data from 150 bp Illumina paired-end reads were generated for *Z. zerumbet* and *Z. montanum*, respectively, and 0.85 M and 0.98 M clean data from 8–10 kb subreads were generated from the two species, respectively.

### Chloroplast genome assembly and annotations

First, the clean Illumina reads were assembled using SOAPdenova (version 2.04) with default parameters into principal contigs [32], and all contigs were sorted and joined into a single draft sequence using the Geneious version 11.0.4 software [33]. Next, the BLASR software was used to compare the PacBio clean data with the single draft sequence and to extract the correction and error correction [34]. Next, the corrected PacBio clean data were assembled using Celera Assembler (version 8.0) with default parameters, generating scaffolds [35]. Next, the assembled scaffolds were mapped back to the Illumina clean reads using GapCloser (version 1.12) for gap closing [32]. Finally, the redundant fragment sequences were removed, thereby generating the final assembled chloroplast genomic sequence.

Annotations of the chloroplast genomes were conducted using the online tool DOGMA (Dual Organellar Genome Annotator) [36] with default parameters and checked manually. BLASTn searches of the National Center for Biotechnology Information (NCBI) website were used to identify and confirm both tRNA and rRNA genes. Last, further verification of the

tRNA genes was carried out using tRNAscanSE with default settings [37]. Circular maps of the chloroplast genomes were drawn using OGDRAWv1.3.1 with default parameters and subsequent manual editing [38].

## Codon usage and RNA editing site prediction

Relative synonymous codon usage (RSCU) in protein-coding genes of *Z. montanum* and *Z. zerumbet* was calculated using the MEGA7 software [39]. Amino acid frequency was also calculated and expressed by the percentage of the codons encoding the same amino acid divided by the total codons. RNA editing sites of 21 protein-coding genes from the two species were investigated using the online program Predictive RNA Editor for Plants (PREP) suite (http://prep.unl.edu/) with a cutoff value of 0.8 [40].

## SNPs and indel detection

To develop specific markers for distinguishing *Z. montanum* and *Z. zerumbet*, the whole chloroplast genomes of *Z. montanum* and *Z. zerumbet* were aligned using the MUMmer software [41] and adjusted manually where necessary using Se-Al 2.0 [42]. The *Z. montanum* chloroplast genome was used as the reference for the SNP and indel analyses.

## SSRs and long repeat analyses of four *Zingiber* species

SSRs of the four *Zingiber*s chloroplast genomes, including *Z. montanum*, *Z. zerumbet*, *Z. officinale* and *Z. spectabile*, were identified using MIcroSAtellite (MISA) (http://pgrc.ipk-gatersleben.de/misa/) [43] with the following settings: 8 for mono-, 5 for di-, 4 for tri-, and 3 for tetra-, penta-, and hexa-nucleotide repeat motifs. The online REPuter software [44] was used to establish the size and location of long repeat sequences, including forward, palindrome, reverse and complement repeat units in the four *Zingiber* chloroplast genomes. The minimal repeat size was set as 30 bp with a repeat identity of 90% and a Hamming distance of 3.

## Sequence divergence analyses of the four *Zingiber* species

To compare the chloroplast genome of *Z. montanum* with three other *Zingiber* species (*Z. zerumbet*, *Z. officinale* and *Z. spectabile*), the mVISTA tool in Shuffle-LAGAN mode [45] was performed using the annotated chloroplast genome of *Z. montanum* as the reference. To detect the variation in the boundaries between the IR and SC regions of the four *Zingiber* chloroplast genomes, the four *Zingiber* chloroplast genomes were compared and analyzed. The nucleotide variability (Pi) among the four whole *Zingiber* chloroplast genomes was calculated using DnaSP version 5.1 [46] with the following settings: window length of 600 bp and step size of 200 bp.

## Selection pressure analysis of the four *Zingiber* species

To estimate selection pressures, nonsynonymous (Ka) and synonymous (Ks) substitution rates of protein-coding genes between the chloroplast genomes of *Z. montanum* and the other three *Zingiber* species (*Z. zerumbet*, *Z. spectabile* and *Z. officinale*) were calculated. The Ka/Ks values for each protein-coding gene were estimated by the KaKs_Calculator [47] with default parameters.

## Phylogeny in the genus *Zingiber* and family Zingiberaceae

In this study, a total of 29 whole chloroplast genome sequences were downloaded from the NCBI database to determine the phylogenetic positions of *Z. montanum* and *Z. zerumbet* in

the genus *Zingiber* and family Zingiberaceae. *Costus pulverulentus*, *Costus viridis* and *Canna indica* were used as outgroups of the family Zingiberaceae. A phylogenetic tree was constructed based on the population SNP matrix of the studied plants, which was obtained using a previously described method [16, 17]. Maximum likelihood (ML) analysis based on the nucleotide substitution model of Tamura-Nei was conducted to construct the phylogenetic tree with MEGA7 software [39]. The ML analysis was performed with 1000 bootstrap replicates.

## Results and discussion

### Chloroplast genome features of *Z. montanum* and *Z. zerumbet*

The raw Illumina and PacBio chloroplast sequencing data had been submitted to the NCBI with SRA numbers SRR8185396 and SRR8184511 for *Z. montanum*, respectively, and SRA numbers SRR8185094 and SRR8184512 for *Z. zerumbet*, respectively. All of these raw data were in the bioproject PRJNA498576. The two whole chloroplast genome sequences had been submitted to GenBank under accession numbers MK262727 and MK262726 for *Z. montanum* and *Z. zerumbet*, respectively. The *Z. montanum* and *Z. zerumbet* chloroplast genomes were 164,464 bp and 163,589 bp in length, respectively (Fig 1). Similar to most other angiosperms, the two genomes had typical quadripartite structure circle molecules consisting of a LSC of 87,856 bp in *Z. montanum* and 89,161 bp in *Z. zerumbet*, a SSC region of 15,803 bp in *Z. montanum* and 15,642 bp in *Z. zerumbet*, and two IR regions of 30,356 bp and 30,449 bp in *Z. montanum* and each 29,393 bp in *Z. zerumbet* (Fig 1 and Table 1). The overall GC contents in the chloroplast genomes of *Z. montanum* and *Z. zerumbet* were 35.75% and 36.27%, respectively (Table 1 and S1 Table). Additionally, the GC contents of the two species were the highest (40.46%-41.02%) in the IR regions, the lowest (29.24%-29.64%) in the SSC regions, and moderate (33.63%-34.31%) in the LSC regions (Table 1), which were similar to the chloroplast genomes of other reported species in the family Zingiberaceae [15–26]. Approximately 50.76%-51.37% of the two *Zingiber* species chloroplast genomes consisted of protein-coding genes (83,496 bp in *Z. montanum* and 84,042 bp in *Z. zerumbet*), 1.74%-1.75% of tRNAs (2,876 bp *Z. montanum* and 2,877 bp in *Z. zerumbet*), and 5.50%-5.52% of rRNAs (9,046 bp in *Z. montanum* and 9,046 bp in *Z. zerumbet*) (S1 Table). For the protein-coding genes, the AT contents of the first, second, and third codons were 55.57%, 62.99%, and 71.26% in *Z. montanum*, respectively, and 55.35%, 62.61%, and 71.20% in *Z. zerumbet*, respectively (S1 Table).

We detected a total of 141 functional genes consisting of 87 protein-coding genes, 46 tRNAs, and eight rRNAs in the *Z. montanum* and *Z. zerumbet* chloroplast genomes, which included 111 unique genes (Tables 1 and 2). Among the 111 unique genes, there were 79 protein-coding genes, 28 tRNAs and four rRNAs in the chloroplast genomes of the two *Zingiber* species (Table 1). Of the protein-coding genes in the *Z. montanum* and *Z. zerumbet* chloroplast genomes, 61 genes were located in the LSC region, 12 genes were in the SSC region and 8 genes were duplicated in the IR regions (Table 1). Eight complete chloroplast genomes, those of *Z. officinale*, *Kaempferia galanga*, *Kaempferia elegans*, *Curcuma zedoaria*, *Curcuma longa*, *Hedychium coronarium*, *Stahlianthus involucratus*, and *Amomum villosum*, belonging to six different genera in the family Zingiberaceae were selected for comparisons with *Z. montanum* and *Z. zerumbet* (Table 1). As shown in Table 1, the *Z. zerumbet* chloroplast genome had the highest GC content (36.27%), while the *Z. montanum* chloroplast genome had the lowest GC content (35.75%). Interestingly, the two IR regions in *Z. zerumbet* (each 29,393 bp) were the shortest, whereas the two IR regions in *Z. montanum* (30,356 bp and 30,449 bp) were the longest (Table 1). There were no significant variations in the numbers of unique total genes, unique protein-coding genes, unique tRNAs and unique rRNAs observed in comparisons of the two *Zingiber* chloroplast genomes with those of the other eight selected chloroplast genome sequences (Table 1).

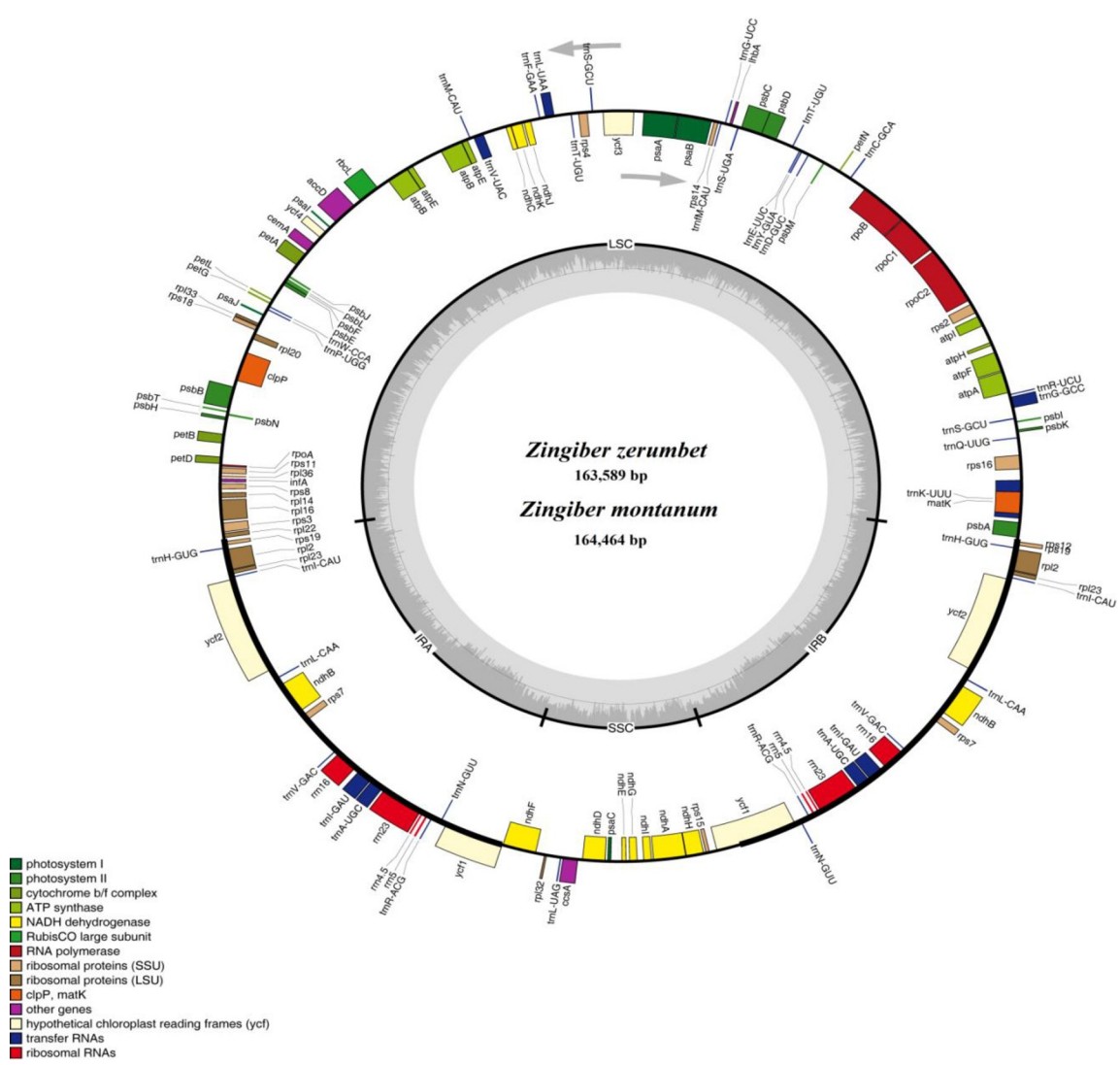

**Fig 1. Circular gene map of the chloroplast genomes of two *Zingiber* species.** The gray arrowheads indicate the direction of the genes. Genes shown inside the circle are transcribed clockwise, and those outside the circle are transcribed counterclockwise. Different genes are color coded. The innermost darker gray corresponds to GC content, whereas the lighter gray corresponds to AT content. IR, inverted repeat; LSC, large single copy region; SSC, small single copy region.

A total of 20 genes were duplicated in the IR regions, including eight protein-coding genes (*ndhB*, *rpl2*, *rpl23*, *rps7*, *rps12*, *rps19*, *ycf1* and *ycf2*), eight tRNA genes (*trnH-GUG*, *trnI-CAU*, *trnL-CAA*, *trnV-GAC*, *trnI-GAU*, *trnA-UGC*, *trnR-ACG* and *trnN-GUU*), and all four rRNAs (*rrn4.5*, *rrn5*, *rrn16* and *rrn23*) (Fig 1 and S2 Table). Seventeen genes (*trnA-UGC*, *trnI-GAU*, *trnG-GCC*, *trnK-UUU*, *trnL-UAA*, *trnV-UAC*, *accD*, *atpF*, *ndhA*, *ndhB*, *rpoC1*, *petB*, *petD*, *rpl2*, *rpl16*, *rps12* and *rps16*) contained one intron, while *ycf3* and *clpP* each contained two introns (S3 Table). Among the 19 intron-containing genes, 4 genes (*trnA-UAC*, *trnI-GAU*, *rpl2* and *ndhB*) occurred in both IRs, 13 genes (*trnG-GCC*, *trnK-UUU*, *trnL-UAA*, *trnV-UAC*, *atpF*, *accD*, *rpoC1*, *petB*, *petD*, *rpl16*, *rps16*, *ycf3* and *clpP*) were distributed in the LSC, one gene (*ndhA*) was in the SSC, and one gene (*rps12*) had its first exon in the LSC and the other two exons in both IRs (Fig 1 and S3 Table). In addition, the *Z. montanum* and *Z. zerumbet* chloroplast genomes had the longest introns of *trnK-UUU* (2,683 bp and 2,606 bp, respectively), all of which were included in the coding region of *matK* (S2 and S3 Tables).

**Table 1. Characteristics of the chloroplast genomes of ten Zingiberaceae species.**

| Genome characteristics | *Zingiber montanum* | *Zingiber zerumbet* | *Zingiber officinale* | *Kaempferia galanga* | *Kaempferia elegans* | *Curcuma zedoaria* | *Curcuma longa* | *Hedychium coronarium* | *Stahlianthus involucratus* | *Amomum villosum* |
|---|---|---|---|---|---|---|---|---|---|---|
| GenBank number | MK262727 | MK262726 | NC_044775 | MK209001 | MK209002 | MK262734 | MK262732 | MK262736 | MK262725 | MK262730 |
| Genome size (bp) | 164,464 | 163,589 | 162,621 | 163,811 | 163,555 | 162,135 | 162,176 | 163,949 | 163,300 | 163,608 |
| LSC length (bp) | 87,856 | 89,161 | 87,486 | 88,405 | 88,020 | 86,966 | 86,984 | 88,581 | 87,498 | 88,680 |
| SSC length (bp) | 15,803 | 15,642 | 15,577 | 15,812 | 15,989 | 15,737 | 15,694 | 15,808 | 15,568 | 15,288 |
| IR length (bp) | 30,356/ 30,449 | 29,393 | 29,779 | 29,797 | 29,773 | 29,716 | 29,749 | 29,780 | 30,117 | 29,820 |
| Total genes (unique) | 141(111) | 141(111) | 133(113) | 133(111) | 133(113) | 141 (111) | 141 (111) | 141(111) | 141(111) | 133(111) |
| CDS (unique) | 87(79) | 87(79) | 87(79) | 87(79) | 87(79) | 87 (79) | 87 (79) | 87(79) | 87(79) | 87(79) |
| tRNA genes (unique) | 46(28) | 46(28) | 38(30) | 38(28) | 38(30) | 46 (28) | 46 (28) | 46(28) | 46(28) | 38(28) |
| rRNA genes (unique) | 8 (4) | 8 (4) | 8 (4) | 8 (4) | 8 (4) | 8 (4) | 8 (4) | 8 (4) | 8 (4) | 8 (4) |
| GC content (%) | | | | | | | | | | |
| Genome | 35.75 | 36.27 | 36.10 | 36.10 | 36.10 | 36.20 | 36.21 | 36.09 | 36.00 | 36.08 |
| CDS | 36.72 | 36.95 | 37.10 | 36.90 | 37.20 | 36.94 | 36.92 | 36.96 | 36.85 | 36.91 |
| LSC | 33.63 | 34.31 | 33.80 | 33.90 | 33.90 | 34.02 | 34.00 | 33.85 | 33.78 | 33.71 |
| SSC | 29.24 | 29.64 | 29.70 | 29.50 | 29.40 | 29.60 | 29.66 | 29.53 | 29.59 | 30.06 |
| IR | 40.51/40.46 | 41.02 | 41.10 | 41.00 | 41.10 | 41.14 | 41.16 | 41.15 | 40.89 | 41.14 |
| Genes with introns | 19 | 19 | 20 | 18 | 17 | 18 | 18 | 18 | 17 | 18 |
| CDS in LSC | 61 | 61 | 60 | 61 | 61 | 61 | 61 | 61 | 61 | 61 |
| CDS in SSC | 12 | 12 | 11 | 12 | 12 | 12 | 12 | 12 | 12 | 12 |
| CDS in IRa | 8 | 8 | 8 | 8 | 8 | 8 | 8 | 8 | 8 | 8 |
| CDS in IRb | 8 | 8 | 8 | 8 | 8 | 8 | 8 | 8 | 8 | 8 |
| Genes in IRs (unique) | 40(20) | 40(20) | 40(20) | 40(20) | 40(20) | 40(20) | 40(20) | 40(20) | 40(20) | 40(20) |

CDS, protein-coding genes; LSC, large single copy region; SSC, small single copy region; IR, inverted repeat.

## Codon usage and predicted RNA editing site analyses

All chloroplast protein-coding genes from *Z. montanum* and *Z. zerumbet* were encoded by 27,832 codons and 28,014 codons, respectively. Similar to most reported Zingiberaceae plants [15–18, 20–21], leucine (Leu) was the most prevalent amino acid in the chloroplast genomes of *Z. montanum* (2888, 10.37%) and *Z. zerumbet* (2896, 10.33%). Conversely, cysteine (Cys), which contained 320 codons in *Z. montanum* (1.14%) and 309 codons in *Z. zerumbet* (1.10%), was the least frequent amino acid in the chloroplast genomes of these two *Zingiber* species (Fig 2 and S4 Table). In the chloroplast genes of the two *Zingiber* species, thirty codons with RSCU>1 were all A/T-ending codons, except for one codon (UUG) that coded for *trnL-CAA* (S4 Table). Stop codon usage was found to be biased toward TAA (RSCU>1.00). Two amino acids, methionine (Met) and tryptophan (Trp), showed no codon bias with RSCU values of 1.00 (S4 Table).

A total of 51 editing sites were identified in 21 protein-coding genes from *Z. montanum* and 19 protein-coding genes from *Z. zerumbet* (Fig 3 and S5 Table). In the *Z. montanum* and

**Table 2. Genes present in the chloroplast genomes of *Z. montanum* and *Z. zerumbet*.**

| Category | Function | Genes |
|---|---|---|
| **Photosynthesis** | Photosystem I | *psaA, psaB, psaC, psaI, psaJ* |
| | Photosystem II | *psbA, psbB, psbC, psbD, psbE, psbF, psbH, psbI, psbJ, psbK, psbL, psbM, psbN, psbT, lhbA* |
| | Cytochrome b/f | *petA, petB\*, petD\*, petG, petL, petN* |
| | ATP synthase | *atpA, atpB, atpE, atpF\*, atpH, atpI* |
| | NADH dehydrogenase | *ndhA\*, ndhB(×2)\*, ndhC, ndhD, ndhE, ndhF, ndhG, ndhH, ndhI, ndhJ, ndhK* |
| | Rubisco | *rbcL* |
| **Self-replication** | RNA polymerase | *rpoA, rpoB, rpoC1\*, rpoC2* |
| | Large subunit ribosomal proteins | *rpl2(×2)\*, rpl14, rpl16\*, rpl20, rpl22, rpl23(×2), rpl32, rpl33, rpl36* |
| | Small subunit ribosomal proteins | *rps2, rps3, rps4, rps7(×2), rps8, rps11, rps12(×2)\*, rps14, rps15, rps16\*, rps18, rps19(×2)* |
| | Ribosomal RNAs | *rrn4.5(×2), rrn5(×2), rrn16(×2), rrn23(×2)* |
| | Transfer RNAs | *trnA-UGC (×4)\*, trnC-GCA, trnD-GUC, trnE-UUC, trnF-GAA, trnfM-CAU, trnG-GCC (×2)\*, trnG-UCC, trnH-GUG (×2), trnI-CAU (×2), trnI-GAU (×4)\*, trnK-UUU (×2)\*, trnL-CAA (×2), trnL-UAA (×2)\*, trnL-UAG, trnM-CAU, trnN-GUU (×2), trnP-UGG, trnQ-UUG, trnR-ACG (×2), trnR-UCU, trnS-GCU (×2), trnS-UGA, trnT-UGU (×2), trnV-GAC (×2), trnV-UAC (×2)\*, trnW-CCA, trnY-GUA* |
| **Others** | Other proteins | *accD\*, ccsA, cemA, clpP\*\*, infA, matK* |
| | Proteins of unknown function | *ycf1(×2), ycf2(×2), ycf3\*\*, ycf4* |

×2, Gene with two copies; ×4, Gene with four copies

\*, Genes containing one intron

\*\*, Genes containing two introns.

*Z. zerumbet* chloroplast genomes that we sequenced, the *ndhB* gene had the highest number of potential editing sites (10, 10), followed by *accD* (3, 6), *matK* (4, 4), *rpoB* (4, 4) and *ycf3* (4, 4) (Fig 3 and S5 Table). Similar to other reported species, such as two *Kaempferia* species [16] and three *Alpinia* species [17], the *ndhB* gene contained the highest number of editing sites. Of these editing sites, all were C-to-T transitions and occurred at the codon first or second positions (S5 Table). In addition, most RNA editing sites in both species led to hydrophobic amino acids, such as leucine (Leu, L), isoleucine (Ile, I), tryptophan (Trp, W), tyrosine (Tyr, Y), valine (Val, V), methionine (Met, M), and phenylalanine (Phe, F) (S5 Table). Similar RNA editing results have already been revealed by previous reports [16, 17].

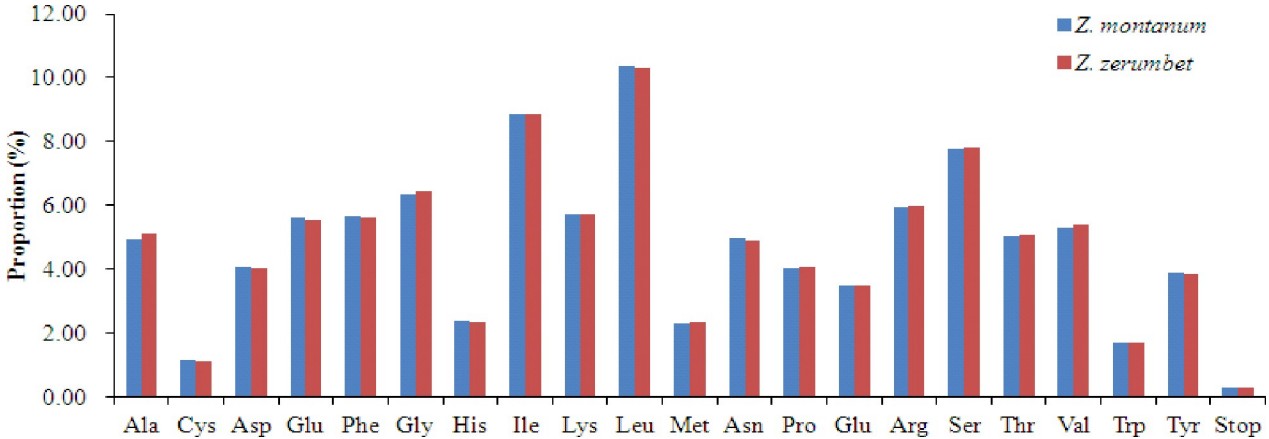

**Fig 2. Amino acid proportion in *Z. montanum* and *Z. zerumbet* protein-coding sequences.**

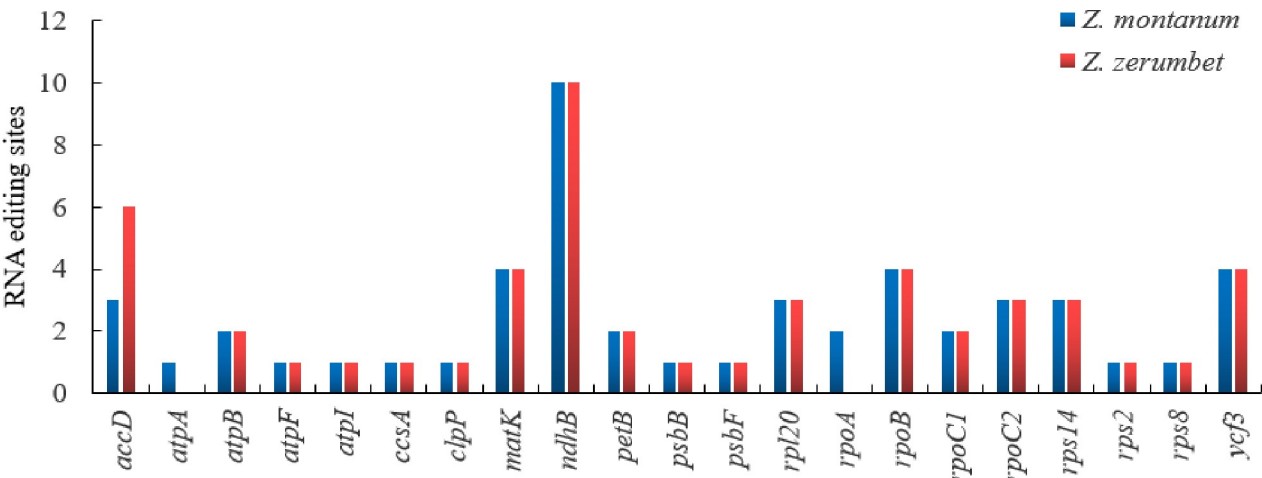

**Fig 3. Predicted RNA editing sites of protein-coding genes in the chloroplast genomes of *Z. montanum* and *Z. zerumbet*.**

## SNP and indel detection between *Z. montanum* and *Z. zerumbet*

Using the *Z. montanum* chloroplast genome as the reference, we compared the SNP/indel loci of the chloroplast genome of *Z. zerumbet*. Two hundred thirty-eight and 251 SNP markers were detected between *Z. montanum* and *Z. zerumbet* in protein-coding genes and intergenic regions, respectively (S6 Table). SNP markers were detected in 49 protein-coding genes in the chloroplast genome of *Z. zerumbet* (Fig 4A and S6 Table). There were 90 synonymous and 148 nonsynonymous SNPs in the protein-coding genes of the *Z. zerumbet* chloroplast genome (S6

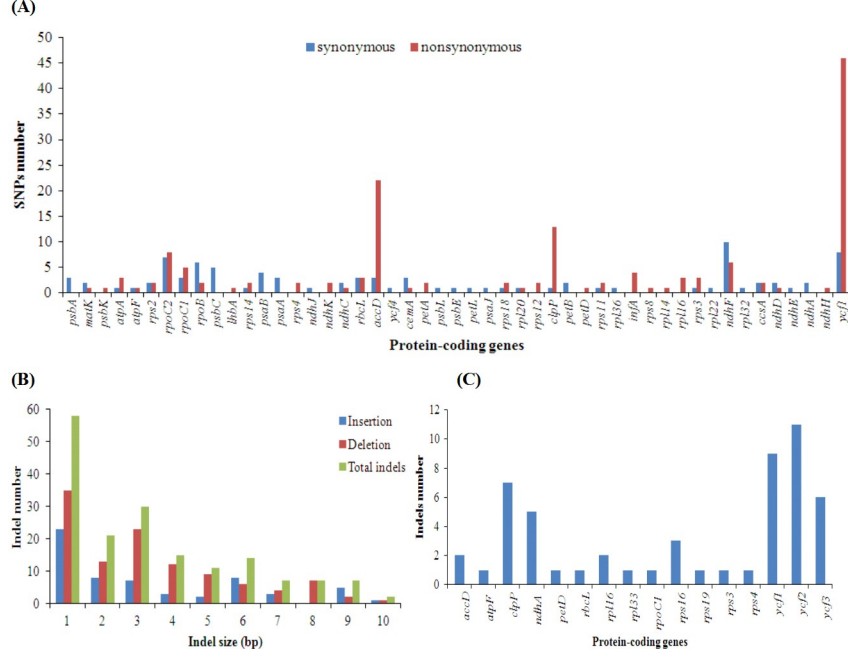

**Fig 4. SNP and indel statistics for the *Z. zerumbet* chloroplast genome.** The *Z. montamum* chloroplast genome was used as the reference sequence for SNP and indel analyses. (A) Synonymous and nonsynonymous SNPs belonging to different protein-coding genes. The genes with zero SNP were not shown. (B) Insertion, deletion and total indel statistics. (C) Indels belonging to different protein-coding genes.

Table). Sixty insertions and 112 deletions were detected between the *Z. montanum* and *Z. zerumbet* chloroplast genomes, respectively (Fig 4B and S7 Table). Sixteen protein-coding genes from the *Z. zerumbet* chloroplast genome contained indels, including *accD*, *atpF*, *clpP*, *ndhA*, *petD*, *rbcL*, *rpl16*, *rpl33*, *rpoC1*, *rps16*, *rps19*, *rps3*, *rps4*, *ycf1*, *ycf2* and *ycf3* (Fig 4C). These results indicated that there were more nucleotide substitutions than between *Alpinia* species but fewer than observed for *Kaempferia* species in the family Zingiberaceae. Comparative analyses of chloroplast genomes revealed 304 SNPs between *Alpinia pumila* and *A. katsumadai*, 367 SNPs between *A. pumila* and *A. oxyphylla* sampled from Guangdong, 331 SNPs between *A. pumila* and *A. zerumbet*, 371 SNPs between *A. pumila* and *A. oxyphylla* sampled from Hainan [17], and 536 SNPs between *K. galanga* and *K. elegans* [16]. By comparison, there were more indels in the two *Zingiber* species than in two *Kaempferia* species and three *Alpinia* species [16, 17]. There were 107 indels between *K. galanga* and *K. elegans* [16], 118 indels between *A. pumila* and *A. katsumadai*, 122 indels between *A. pumila* and *A. oxyphylla* sampled from Guangdong, 115 indels between *A. pumila* and *A. zerumbet*, and 120 indels between *A. pumila* and *A. oxyphylla* sampled from Hainan [17]. The SNP and indel resources produced in this study could be used for phylogenetic analysis and species identification in the genus *Zingiber* and family Zingiberaceae in the future.

## SSR and long repeat analyses

SSRs, with a repeat unit length ranging from one to six nucleotides or more, are widely distributed in chloroplast genomes [15–18, 21]. A total of 240, 200, 190 and 197 SSRs were detected in the chloroplast genomes of *Z. montanum*, *Z. zerumbet*, *Z. spectabile*, and *Z. officinale*, respectively (Fig 5A and S8 Table). Among these SSRs, the noncoding region had the most SSRs (129–169 loci, 64.50%-70.41%), whereas the coding region had the fewest SSRs (59–71

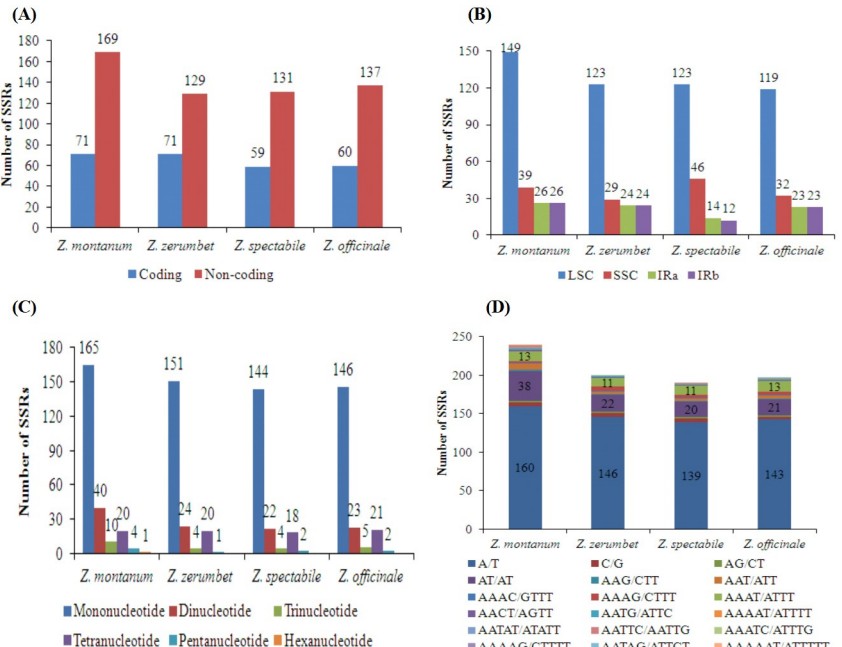

**Fig 5. Comparison of simple sequence repeats among four chloroplast genomes of *Zingiber* species.** (A) SSRs distribution between coding and noncoding regions detected in the four *Zingiber* species chloroplast genomes. (B) Frequencies of identified SSRs in LSC, SSC and IR regions. (C) Number of different SSR types detected in four *Zingiber* species chloroplast genomes. (D) Frequency of identified SSRs in different repeat class types.

loci, 29.59%-35.50%) (Fig 5A). The majority of SSRs were located in the LSC regions (119–149 loci, 60.40%-64.73%); only a small portion were located in the SSC regions (29–46 loci, 14.50%-24.21%) and IR regions (12–26 loci, 6.31%-11.67%) of the four *Zingiber* chloroplast genomes (Fig 5B). Mono-, di-, tri-, tetra-, and penta-nucleotide SSRs were all detected in the four chloroplast genomes (Fig 5C). Additionally, only one hexanucleotide SSR was detected in the chloroplast genome of *Z. montanum* (Fig 5C). Among the different types of SSRs, mono-nucleotide repeats were the most abundant, accounting for 68.75%-75.78% of all SSRs, followed by dinucleotide (11.57%-16.66%) and tetranucleotide (8.33%-10.65%) repeats (Fig 5D and S8 Table). Mononucleotide SSRs were especially rich in A/T repeats (96.52%-97.94%) among the four *Zingiber* chloroplast genomes (Fig 5D). These results were consistent with most reported Zingiberaceae species [15–18, 21]. The second most abundant SSR types were AT/AT repeats, which were the majority of dinucleotide repeats (90.90%-95.00%). AAAT/ATTT repeats were the third most abundant SSR types in the four chloroplast genomes (55.00%-65.00%) (Fig 5D).

We also analyzed long repeats by REPuter and found the following four categories of long repeats: palindromic, forward, reverse, and complement. A total of 176 long repeats were found among the four chloroplast genomes. In detail, there were 50 (24 palindromic and 26 forward), 50 (9 palindromic, 37 forward, 3 reverse and 1 complement), 34 (19 palindromic, 14 forward and 1 reverse) and 42 (18 palindromic, 19 forward, 4 reverse, and 1 complement) long repeats in *Z. montanum*, *Z. zerumbet*, *Z. spectabile* and *Z. officinale*, respectively (Fig 6A and S9 Table). Interestingly, there were no complement repeats in the chloroplast genomes of *Z. montanum* and *Z. spectabile* (Fig 6A). With 24 palindromic repeats, *Z. montanum* contained the highest number of palindromic repeats, while *Z. zerumbet* contained the highest number of forward repeats at 37; *Z. officinale* contained 4 reverse repeats, the highest among the four compared chloroplast genomes (Fig 6B–6D). Palindromic and forward repeats measuring > 60 bp were found to be the most common in the chloroplast genome of *Z. montanum* (Fig 6B and 6C). Conversely, 30–60 bp palindromic and forward repeats were the most common in

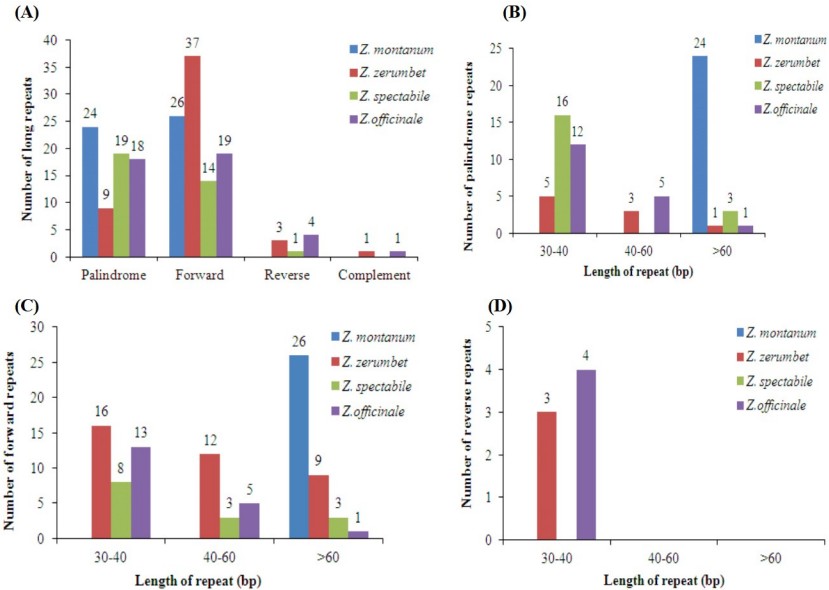

**Fig 6. Analysis of long repeat sequences in the chloroplast genomes of the four *Zingiber* species.** (A) Total of four long repeat types; (B) frequency of palindromic repeats by length; (C) frequency of forward repeats by length; and (D) frequency of reverse repeats by length.

the other three chloroplast genomes (Fig 6B and 6C). Furthermore, almost all of the reverse repeats were less than 60 bp in the four chloroplast genomes (Fig 6D).

## Comparative genomic analysis

The whole chloroplast genomes of the two sequenced *Zingiber* species and two published *Zingiber* species were compared using mVISTA, with *Z. montanum* being used as the reference (Fig 7). The mVISTA results indicated that the LSC and SSC regions were more divergent than the two IR regions. This phenomenon also occurred in most land plants [15–18]. The divergence level of the noncoding regions was higher than that of the coding regions. Approximately 13 highly divergent regions were found in mVISTA, and they were mainly distributed in noncoding regions, including start-*psbA*, *trnfM-CAU-rps14*, *ycf1-ndhF*, *rbcL-accD*, *accD-psaI*, *atpI-atpH*, *ccsA-ndhD*, *rps18-rpl20*, and *trnE-UUC-trnT-UGU*, and in 4 genes, namely, *ycf1*, *ycf2*, *accD*, and *rps19* (Fig 7). Among these regions, *accD-psaI*, *atpI-atpH*, *ccsA-ndhD*, *trnE-UUC-trnT-UGU*, *ycf1*, and *ycf2* have also been observed in other Zingiberaceae plant chloroplast genomes [15–18, 20]. Furthermore, the four junctions of LSC/IRa, LSC/IRb, SSC/

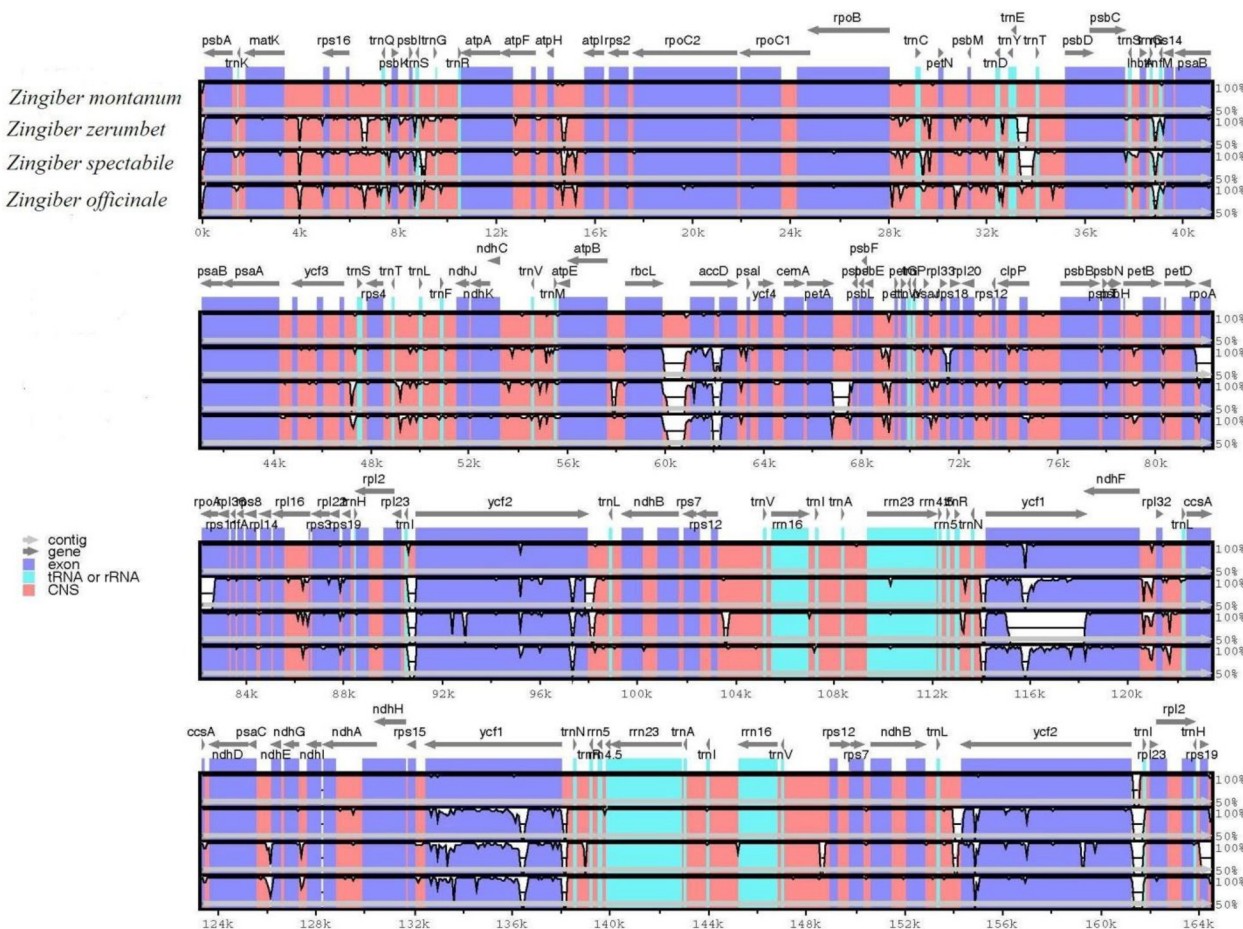

**Fig 7. Sequence alignment of the four *Zingiber* chloroplast genomes in mVISTA.** The chloroplast genome of *Z. montanum* was used as a reference. Gray arrows and thick black lines above the alignment indicate gene orientation. Purple bars represent exons, sky-blue bars represent transfer RNA (tRNA) and ribosomal RNA (rRNA) and red bars represent noncoding sequences (CNS). The horizontal axis indicates the coordinates within the chloroplast genome. The vertical scale represents the identity percentage ranging from 50% to 100%. White represents regions with sequence variation among the four species.

IRa and SSC/IRb for the four *Zingiber* chloroplast genomes are shown in a detailed comparison (S1 Fig). In the four junctions, the genes in the border regions, including *rpl22*, *rps19*, Ψ*ycf1*, *ndhF*, *ycf1*, *rps19*, and *psbA*, were the same in *Z. montanum*, *Z. zerumbet*, and *Z. officinale*. However, in *Z. spectabile*, the *trnM- ycf2* sequence was located in the junctions of the LSC/IRa region, which was missing the *rpl22* and *rps19* genes. The *trnH* gene was at one end of the IRb region in *Z. spectabile* instead of the *rps19* gene in the LSC/IRb junction.

Moreover, the four *Zingiber* species were detected to have highly divergent regions in their chloroplast genomes using DnaSP by sliding window analysis (Fig 8). Among the 85 protein-coding regions (CDS), nucleotide diversity (Pi) values ranged from 0.0006 (*atpI*) to 0.2394 (*rps19*) and had an average value of 0.0084. Three protein-coding regions (*ycf1*, *trnfM-CAU*, and *rps19*) showed remarkably high values (Pi>0.02; Fig 8A and S10 Table). For the 128 non-coding regions, Pi values ranged from 0.00069 (*rpoC1*-CDS2-*rpoC1*-CDS1) to 0.2777 (*ycf1-ndhF*) and had an average of 0.01406. These results also proved that the average value of Pi in the noncoding regions was more than 1.5 times that in the coding regions. Sixteen of these regions had remarkably high values (Pi>0.0215), including *rps18-rpl20*, *accD-psaI*, *psaC-ndhE*, *psbA-trnK-UUU*, *trnfM-CAU-rps14*, *trnE-UUC-trnT-UGU*, *ccsA-ndhD*, *psbC-trnS-UGA*, *start-psbA*, *petA-psbJ*, *rbcL-accD*, *ycf2-trnI-CAU*, *accD-CDS1-accD-CDS2*, *trnI-CAU-ycf2*, *psbT-psbN* and *ycf1-ndhF* (Fig 8B and S10 Table). However, for the selection of effective and useful markers, both the length and Pi values of the highly variable regions must be considered. Among the nineteen regions, six regions (*trnfM-CAU*, *accD-CDS1-accD-CDS2*, *ycf2-trnI-CAU*, *trnI-CAU-ycf2*, *psbT-psbN* and *ycf1-ndhF*) were too short to be used as molecular markers. Finally, the other thirteen highly divergent regions could be suitable DNA markers for species identification in the genus *Zingiber*.

A) Coding region

B) Noncoding region

**Fig 8. Sliding window analysis of the whole chloroplast genomes among four *Zingiber* species.** Window length: 800 bp; step size: 200 bp. X-axis: position of the window midpoint.

## Selection events in unique protein-coding genes

The Ka/Ks ratio is useful for measuring selection pressure on a specific gene [48–50]. In most cases, the Ka/Ks ratio is less than 1, indicating a purifying selection; when Ka/Ks = 1, it reveals a neutral selection; and if Ka/Ks>1, it means a positive selection on the specific gene [48–50]. In this study, we compared the Ka/Ks ratios of 78 shared unique protein-coding genes in the *Z. montanum* chloroplast genome and the chloroplast genomes of the following three other related *Zingiber* species: *Z. officinale*, *Z. spectabile*, and *Z. zerumbet* (S2 Fig). The results indicated that the Ka/Ks values of some genes were NA or 50. These phenomena values occurred when the Ks values were notably low or the two aligned sequences exhibited 100% perfect matches. In these circumstances, we replaced NA or 50 with 0. As a result, *ATP synthase* (*atpA* and *atpB*), *RNA polymerase* (*rpoA*), *small subunit ribosomal protein* (*rps3*) and other protein-coding genes (*accD*, *clpP*, *ycf1*, and *ycf2*) with Ka/Ks>1 were detected, indicating that these genes were undergoing positive selection (S2 Fig). Moreover, the Ka/Ks ratios of three genes (*clpP*, *ycf1* and *ycf2*) in three pairwise comparisons of *Z. montanum-Z. officinale*, *Z. montanum-Z. spectabile*, and *Z. montanum-Z. zerumbet*, respectively, were all >1, indicating that the three genes *clpP*, *ycf1* and *ycf2* exhibited critical adaptation evolution to diverse environments.

## Inferring phylogeny in the genus *Zingiber* and family Zingiberaceae

The chloroplast genome sequences provided useful genomic resources for phylogenetic studies [51, 52]. Several previous studies have successfully used protein-coding genes, whole chloroplast genome sequences, or chloroplast SNP-based matrices for phylogenetic inference in the family Zingiberaceae [13, 15–26]. In the present study, a phylogenetic tree was reconstructed with a chloroplast SNP matrix from 31 chloroplast genomes using the ML method with *C. pulverulentus*, *C. viridis* and *C. indica* as outgroups. As shown in Fig 9, plants belonging to six genera from the family Zingiberaceae were basically divided into the following two clusters with

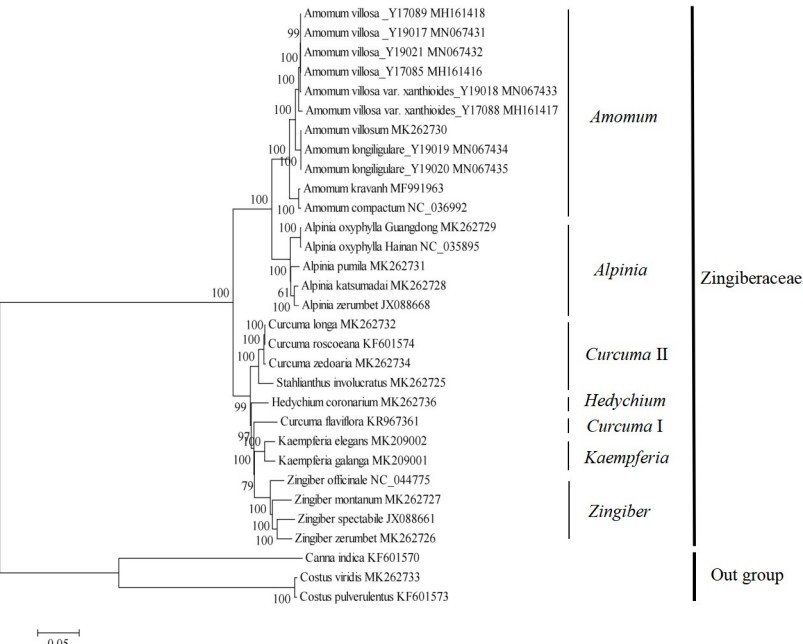

**Fig 9. Phylogenetic relationships constructed with SNPs from 31 chloroplast genomes using the maximum likelihood method.** The bootstrap values were based on 1,000 replicates and are indicated next to the branches.

high bootstrap values of 100%: one included two genera, *Amomum* and *Alpinia*, and the other included four genera, *Curcuma*, *Hedychium*, *Kaempferia* and *Zingiber*. The chloroplast SNP-based phylogeny analyses also showed that *Zingiber* was a monophyletic genus that was sister to the genus *Kaempferia* with moderate bootstrap values of 79% (Fig 9). In the genus *Zingiber*, *Z. spectabile* and *Z. zerumbet* were grouped in a sister branch with high bootstrap values of 100% and then clustered step by step with *Z. montanum* and *Z. officinale* with high bootstrap values of 100% (Fig 9). Interestingly, *Z. zerumbet* first grouped with *Z. spectabile*, rather than *Z. montanum*. Nevertheless, our molecular phylogeny analyses were congruent with a previous AFLP-based DNA marker study, which showed that *Z. montanum* and *Z. zerumbet* were phylogenetically closer to each other than to *Z. officinale* [14]. Our findings also confirmed that chloroplast SNPs were useful resources for phylogenetic analyses in the genus *Zingiber* and family Zingiberaceae.

## Conclusions

We sequenced and analyzed the complete chloroplast genomes of *Z. montanum* and *Z. zerumbet* from the family Zingiberaceae. The genome structures, gene information, amino acid frequencies, codon usage patterns and RNA editing sites of the two *Zingiber* species were determined. Comparative chloroplast genome analyses of *Z. montanum* and *Z. zerumbet* detected 489 SNPs and 172 indels. A total of 827 SSRs and 176 long repeats were identified in four *Zingiber* species chloroplast genomes. Thirteen divergent regions (*ycf1*, *rps19*, *rps18-rpl20*, *accD-psaI*, *psaC-ndhE*, *psbA-trnK-UUU*, *trnfM-CAU-rps14*, *trnE-UUC-trnT-UGU*, *ccsA-ndhD*, *psbC-trnS-UGA*, *start-psbA*, *petA-psbJ*, and *rbcL-accD*) were identified and might be useful for future species identification and phylogeny analysis in the genus *Zingiber*. Selection pressure analysis in the genus *Zingiber* indicated that the *atpA*, *atpB*, *rpoA*, *rps3*, *accD*, *clpP*, *ycf1*, and *ycf2* genes were under positive selection. The chloroplast SNP-based phylogeny analyses determined that *Zingiber* was a monophyletic sister branch to *Kaempferia* and that phylogenetic relationships of the four *Zingiber* species could be clearly identified.

## Supporting information

**S1 Table. Features of the chloroplast genomes of *Z. montanum* and *Z. zerumbet*.**
(DOCX)

**S2 Table. The chloroplast genome annotations of two *Zingiber* species.**
(XLSX)

**S3 Table. Genes with introns in the chloroplast genomes of *Z. montanum* and *Z. zerumbet*.**
(DOCX)

**S4 Table. Codon usages of protein-coding genes in the chloroplast genomes of two *Zingiber* species.**
(XLSX)

**S5 Table. RNA editing sites analysis of two *Zingiber* species.**
(XLS)

**S6 Table. SNPs detected between the *Z. montanum* and *Z. zerumbet* chloroplast genomes.**
(XLSX)

**S7 Table. Indels detected between the *Z. montanum* and *Z. zerumbet* chloroplast genomes.**
(XLSX)

**S8 Table. SSRs distribution among four *Zingiber* chloroplast genomes.**
(XLSX)

**S9 Table. Long repeats distribution among four *Zingiber* chloroplast genomes.**
(XLSX)

**S10 Table. Nucleotide diversity values among four *Zingiber* chloroplast genomes.**
(XLSX)

**S1 Fig. Comparison of the borders of the LSC, SSC, and IR regions among four *Zingiber* species chloroplast genomes. Ψ**, pseudogenes. Boxes above the main line indicate the adjacent border genes. The figure is not to scale with respect to sequence length and shows relative changes only at or near the IR/SC borders.
(DOCX)

**S2 Fig. Ka/Ks ratios of 78 protein-coding genes from the *Z. montanum* chloroplast genome vs. three *Zingiber* species.** Ka, nonsynonymous; Ks, synonymous; *Zm*, *Z. montanum*; *Zo*, *Z. officinale*; *Zs*, *Z. spectabile*; *Zz*, *Z. zerumbet*.
(DOCX)

## Author Contributions

**Conceptualization:** Dong-Mei Li.

**Data curation:** Dong-Mei Li.

**Formal analysis:** Dong-Mei Li, Yuan-Jun Ye, Ye-Chun Xu, Jin-Mei Liu.

**Funding acquisition:** Dong-Mei Li.

**Investigation:** Yuan-Jun Ye, Ye-Chun Xu, Jin-Mei Liu.

**Project administration:** Dong-Mei Li.

**Supervision:** Gen-Fa Zhu.

**Writing – original draft:** Dong-Mei Li.

**Writing – review & editing:** Dong-Mei Li.

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
