## [Decision Letter · Decision Letter 0]

5 May 2020

PONE-D-20-07989

Complete chloroplast genomes of Zingiber montanum and Zingiber zerumbet: genome structures, comparative and phylogenetic analyses

PLOS ONE

Dear Dr. Li,

Thank you for submitting your manuscript to PLOS ONE. After careful consideration, we feel that it has merit but does not fully meet PLOS ONE’s publication criteria as it currently stands. Therefore, we invite you to submit a revised version of the manuscript that addresses the points raised during the review process.

We would appreciate receiving your revised manuscript by Jun 19 2020 11:59PM. To enhance the reproducibility of your results, we recommend that if applicable you deposit your laboratory protocols in protocols.io, where a protocol can be assigned its own identifier (DOI) such that it can be cited independently in the future. For instructions see: http://journals.plos.org/plosone/s/submission-guidelines#loc-laboratory-protocols

We look forward to receiving your revised manuscript.

Kind regards,

Tzen-Yuh Chiang

Academic Editor

PLOS ONE

Journal Requirements:

Reviewers' comments:

Reviewer's Responses to Questions

**Comments to the Author**

1. Is the manuscript technically sound, and do the data support the conclusions?

Reviewer #1: Yes

Reviewer #2: Yes

2. Has the statistical analysis been performed appropriately and rigorously? 

Reviewer #1: Yes

Reviewer #2: Yes

3. Have the authors made all data underlying the findings in their manuscript fully available?

Reviewer #1: Yes

Reviewer #2: No

4. Is the manuscript presented in an intelligible fashion and written in standard English?

Reviewer #1: No

Reviewer #2: Yes

5. Review Comments to the Author

Reviewer #1: With the advent of high-throughput sequencing technology, the assembling of complete plastome sequences have become highly affordable in many lab. It is a simple math that complete plastome sequence can provide much more information for phylogenetic reconstruction and species identification. Therefore, it is highly welcome that more plastome sequences can be reported. The research has been executed correctly following the standard protocol and the bioinformatics had been analyzed correctly. However, the overall writing of the manuscript in the introduction is fairly poor, and not in an intelligible fashion in standard English. For example, in the first sentence, the author wrote: all of which are "natively" distributed in tropical to ........ It is interesting that the author used the term "native", as this suggest to me that the author intended to separate native from introduced ranged. In the sentence about the distribution of Zingiber zerumbet, it says "widely distributed in all tropical regions, especially in South Asia and Southeast Asia." This sentence suggest Z. zerumbet is also distributed in tropical America and Africa, as the species is distributed in "all tropical regions." the sentence was followed by ", which is "produced" in Guangdong, Guangxi, Hainan, Yunnan, and Taiwan province in China". I was pondering for the use of "produced" for a while as I have never seen it been used under such a writing context. I assume the author want to state that Z. zerumbet was cultivated and sold in these regions. This is an example of how this manuscript was not written in an intelligible fashion and in standard English. I also have an issue with treating Taiwan as a province of China, as through the current COVID-19 pandemic, it has become crystal clear that Taiwan is an independent country, not part of China. There are also misinterpretations of published journal article. For example, the scope of reference 13 (Kress et al. 2002) was not to identify some Zingiber species. Kress et al. (2002) was a family-level phylogenetic analyses aiming to clarify the classification within the zinger family, not to identify specific species. It is incorrect to cite this literature in supporting the author's statement that "Recently, several studies have also used molecular data to identify some Zingiber species". In the last sentence of the introduction page, the author wrote "but they have been limited in high resolution for interspecific identifications". I simply don't understand what does this sentence mean. In the sentence followed: "Therefore, a more accurate method of plant identification is essential for Zingiber species". If the goal of the article is plant species identification, I don't think the experimental design, i.e., based on plastome sequence of two plants, is adequate.

I hate to judge the value of a manuscript based on the writing but the manuscript simply is not well written. With rapid progress of the field, it is really hard to convinced me that a report with two plastomes and this quality of writing can be published in a journal such as PLoS ONE.

Reviewer #2: In this study, Li et al. reported the complete chloroplast genomes of two species, Zingiber montanum and Z. zerumbet. While descriptive, the analyses are technically sound and the sequences could serve important resources for future study. I would like to remind the authors to also deposit the raw Illumina and PacBio reads onto NCBI in addition to the assembled genome (if they have not already done so). They should group all the raw reads and assembled genomes under the same "bioproject" and report the bioproject number in the manuscript, so that future users can retrieve the raw data and assembled genomes under the same bioproject.

According to online database, Z. zerumbet appears to be an invasive species into Taiwan and is not native. I suggest the authors change the description of its native distribution range in the first paragraph of Introduction.

Here are a few minor comments:

Fig 4: It is unclear why the authors chose specific subsets of genes to present in this figure. For example, in Fig 4c, ycf1, ycf2, and ycf 3 all have high indel number, and one would therefore like to check whether this causes alignment error and is associated with the high SNP number in Fig 4a. However, Fig 4a does not contain ycf2 or ycf3. Are those genes not in the graphs because they have zero SNP or indel?

Fig 5: Is this compared to the Z. montamum genome? Specify.

Fig 6: It would be great to add a graphical explanation of what are palindromic, forward, reverse, and complement.

Fig 7: I don't really see "white peaks", only white valleys.

About Ka/Ks comparison, one should note that when the sequences being compared are relatively similar, the number of synonymous and non-synonymous changes are low. Under such circumstances, the high Ka/Ks ratio might just be created by a few more non-synonymous chances than synonymous ones. In other words, the power to detect positive selection is low in these circumstances even though Ka/Ks > 1. The authors should at least acknowledge this.

6. PLOS authors have the option to publish the peer review history of their article (what does this mean?). If published, this will include your full peer review and any attached files.

Reviewer #1: No

Reviewer #2: No

---

## [Author Response · Author response to Decision Letter 0]

24 May 2020

Reviewer #1: With the advent of high-throughput sequencing technology, the assembling of complete plastome sequences have become highly affordable in many lab. It is a simple math that complete plastome sequence can provide much more information for phylogenetic reconstruction and species identification. Therefore, it is highly welcome that more plastome sequences can be reported. The research has been executed correctly following the standard protocol and the bioinformatics had been analyzed correctly. However, the overall writing of the manuscript in the introduction is fairly poor, and not in an intelligible fashion in standard English. For example, in the first sentence, the author wrote: all of which are "natively" distributed in tropical to ........ It is interesting that the author used the term "native", as this suggest to me that the author intended to separate native from introduced ranged. In the sentence about the distribution of Zingiber zerumbet, it says "widely distributed in all tropical regions, especially in South Asia and Southeast Asia." This sentence suggest Z. zerumbet is also distributed in tropical America and Africa, as the species is distributed in "all tropical regions." the sentence was followed by ", which is "produced" in Guangdong, Guangxi, Hainan, Yunnan, and Taiwan province in China". I was pondering for the use of "produced" for a while as I have never seen it been used under such a writing context. I assume the author want to state that Z. zerumbet was cultivated and sold in these regions. This is an example of how this manuscript was not written in an intelligible fashion and in standard English. 

Answer: Yes, we agree with this good idea and have revised the first paragraph of the Introduction as follows in red:

Zingiber Boehm., belonging to the family Zingiberaceae, consists of between 100 and 150 species, all of which are widely distributed in southern and southeastern Asia, with particular concentrations in Thailand and southern China [1-4]. There are more than 40 Zingiber species in China, among which 13 are reported to have medicinal value [1, 2, 5]. In addition, most species have an assemblage of tightly clasped, overlapping bracts that often age to yellow, red, or chestnut brown and are often highly showy and long-lived, leading to the cultivation of a number of species for landscaping and cut-flower uses [2-4]. Both Zingiber montanum (J. König) A. Dietr and Zingiber zerumbet (Linnaeus) Rosc. ex Smith are useful medicinal and ornamental plants in this genus [2-5]. Z. montanum is endemic to the Guangdong, Guangxi, Hainan and Yunnan provinces of China [4]. Chemical compositions of the Z. montanum rhizome have antidiarrheal, antioxidant, antibacterial, antifungal, allelopathic and acetylcholinesterase inhibitory properties [3, 4, 6-8]. Z. zerumbet, commonly known as “shampoo ginger”, is found across southern China (Guangdong, Guangxi, Hainan, and Yunnan provinces), most of Southeast Asia, Myanmar, India, and Sri Lanka [1-4]. Zerumbone from the Z. zerumbet rhizome has been reported to suppress the phagocytic activity of human neutrophils [9], to prevent and treat tooth decay disease [10], to cure osteoarthritis of the knee [11], and to treat various immune-inflammatory related disorders [12].

I also have an issue with treating Taiwan as a province of China, as through the current COVID-19 pandemic, it has become crystal clear that Taiwan is an independent country, not part of China. 

Answer: This is not an academic question. We declare that there is only one China in the world and that Taiwan is a part of China’s territory.

There are also misinterpretations of published journal article. For example, the scope of reference 13 (Kress et al. 2002) was not to identify some Zingiber species. Kress et al. (2002) was a family-level phylogenetic analyses aiming to clarify the classification within the zinger family, not to identify specific species. It is incorrect to cite this literature in supporting the author's statement that "Recently, several studies have also used molecular data to identify some Zingiber species". In the last sentence of the introduction page, the author wrote "but they have been limited in high resolution for interspecific identifications". I simply don't understand what does this sentence mean. In the sentence followed: "Therefore, a more accurate method of plant identification is essential for Zingiber species". 

Answer: We disagree with this comment. Reference 13 (Kress et al. 2002) is not only a family-level phylogenetic analysis aiming to clarify the classification within the Zingiberaceae family but also identifies some interspecific species at genus-level, such as the Amomum, Alpinia, Curcuma, Hedychium and Zingiber species. In Fig. 8, 9 and 10 from reference 13 (Kress et al. 2002), there is weak resolution and support (bootstrap value <50%) among the six Zingiber species (Zingiber corallinum, Zingiber wrayi, Zingiber sulphureum, Zingiber gramineum, Zingiber ellipticum and Zingiber species) using nuclear internal transcribed spacer (ITS) and chloroplast matK regions. 

If the goal of the article is plant species identification, I don't think the experimental design, i.e., based on plastome sequence of two plants, is adequate.

I hate to judge the value of a manuscript based on the writing but the manuscript simply is not well written. With rapid progress of the field, it is really hard to convinced me that a report with two plastomes and this quality of writing can be published in a journal such as PLoS ONE.

Answer: We have improved the quality of the manuscript with help from American Journal Experts (AJE). We believe that the revised manuscript is readable. 

Reviewer #2: In this study, Li et al. reported the complete chloroplast genomes of two species, Zingiber montanum and Z. zerumbet. While descriptive, the analyses are technically sound and the sequences could serve important resources for future study. I would like to remind the authors to also deposit the raw Illumina and PacBio reads onto NCBI in addition to the assembled genome (if they have not already done so). They should group all the raw reads and assembled genomes under the same "bioproject" and report the bioproject number in the manuscript, so that future users can retrieve the raw data and assembled genomes under the same bioproject.

Answer: Yes, we deposited the raw Illumina and PacBio reads into the NCBI. The Z. montanum chloroplast sequencing data have SRA numbers SRR8185396 and SRR8184511. The Z. zerumbet chloroplast sequencing data have SRA numbers SRR8185094 and SRR8184512. The final assembled chloroplast genomic sequences have been submitted to GenBank under accession numbers MK262727 and MK262726 for Z. montanum and Z. zerumbet, respectively. The bioproject number is PRJNA498576. We have also added the bioproject number in the manuscript.

According to online database, Z. zerumbet appears to be an invasive species into Taiwan and is not native. I suggest the authors change the description of its native distribution range in the first paragraph of Introduction.

Answer: Yes, we agree with this comment and have revised this sentence as follows:

Z. zerumbet, commonly known as the “shampoo ginger”, is found across southern China (Guangdong, Guangxi, Hainan and Yunnan provinces), most of Southeast Asia, Myanmar, India, and Sri Lanka [1-4].

Here are a few minor comments:

Fig 4: It is unclear why the authors chose specific subsets of genes to present in this figure. For example, in Fig 4c, ycf1, ycf2, and ycf 3 all have high indel number, and one would therefore like to check whether this causes alignment error and is associated with the high SNP number in Fig 4a. However, Fig 4a does not contain ycf2 or ycf3. Are those genes not in the graphs because they have zero SNP or indel?

Answer: We checked the SNP and indel results once again through alignment. According to the indel results (Table S7), both ycf2 and ycf3 contain indels. Based on the SNP results (Table S6), both ycf2 and ycf3 have zero synonymous and nonsynonymous SNP. Therefore, Fig. 4a does not contain ycf2 and ycf3 because ycf2 and ycf3 have zero synonymous and nonsynonymous SNPs. 

Fig 5: Is this compared to the Z. montamum genome? Specify.

Answer: Fig. 5 shows the distribution of SSRs among four chloroplast genomes in Zingiber species. First, the SSRs in each chloroplast genome were detected independently. Then, we compared the SSR results among the four chloroplast genomes. Therefore, Fig. 5 is not compared to the Z. montamum genome. We have revised our explanation of Fig. 5. 

Fig. 5. Comparison of simple sequence repeats among four chloroplast genomes of Zingiber species.

Fig 6: It would be great to add a graphical explanation of what are palindromic, forward, reverse, and complement.

Answer: Long repeat sequences include forward, palindrome, reverse and complement repeats. The sizes and locations of the four types of long repeats (forward, palindrome, reverse and complement) were obtained by the online REPuter software [44]. The minimal repeat size was set as 30 bp with a repeat identity of 90% and a Hamming distance of 3. 

 Table S9 explains the sizes and locations of the forward, palindrome, reverse and complement repeats in the chloroplast genomes of the four Zingiber species.

44. Kurtz S, Choudhuri JV, Ohlebusch E, Schleiermacher C, Stoye J, Giegerich R. REPuter: The manifold applications of repeat analysis on a genomic scale. Nucleic Acids Res. 2001; 29: 4633-4642.

Fig 7: I don't really see "white peaks", only white valleys.

Answer: “White peaks” indicates spiky peaks. In fact, regions with sequence variation among the four species included white peaks and white valleys. In our description, the sequence variation regions were not all included. Therefore, we revised the sentence to “White represents regions with sequence variation among the four species.” instead of “white peaks represent differences in chloroplast genomes”. The description of Fig. 7 is as follows:

Fig. 7. Sequence alignment of four Zingiber chloroplast genomes in mVISTA. The chloroplast genome of Z. montanum was used as a reference. Gray arrows and thick black lines above the alignment indicate gene orientation. Purple bars represent exons, sky-blue bars represent transfer RNA (tRNA) and ribosomal RNA (rRNA) and red bars represent non-coding sequences (CNS). The horizontal axis indicates the coordinates within the chloroplast genome. The vertical scale represents the identity percentage ranging from 50% to 100%. White represents regions with sequence variation among the four species. 

About Ka/Ks comparison, one should note that when the sequences being compared are relatively similar, the number of synonymous and non-synonymous changes are low. Under such circumstances, the high Ka/Ks ratio might just be created by a few more non-synonymous chances than synonymous ones. In other words, the power to detect positive selection is low in these circumstances even though Ka/Ks > 1. The authors should at least acknowledge this.

Answer: Yes, we agree with this comment. Our analysis results indicated that the non-synonymous (Ka)/synonymous (Ks) values of some genes were NA or 50. These phenomena values occurred when the Ks values were notably low or the two aligned sequences exhibited 100% perfect matches. In these circumstances, we replaced NA or 50 with 0.

---

## [Decision Letter · Decision Letter 1]

30 Jun 2020

PONE-D-20-07989R1

Complete chloroplast genomes of Zingiber montanum and Zingiber zerumbet: genome structure, comparative and phylogenetic analyses

PLOS ONE

Dear Dr. Li,

Thank you for submitting your manuscript to PLOS ONE. After careful consideration, we feel that it has merit but does not fully meet PLOS ONE’s publication criteria as it currently stands. Therefore, we invite you to submit a revised version of the manuscript that addresses the points raised during the review process.

We look forward to receiving your revised manuscript.

Kind regards,

Tzen-Yuh Chiang

Academic Editor

PLOS ONE

Reviewers' comments:

Reviewer's Responses to Questions

**Comments to the Author**

1. If the authors have adequately addressed your comments raised in a previous round of review and you feel that this manuscript is now acceptable for publication, you may indicate that here to bypass the “Comments to the Author” section, enter your conflict of interest statement in the “Confidential to Editor” section, and submit your "Accept" recommendation.

Reviewer #2: All comments have been addressed

2. Is the manuscript technically sound, and do the data support the conclusions?

Reviewer #2: Yes

3. Has the statistical analysis been performed appropriately and rigorously? 

Reviewer #2: Yes

4. Have the authors made all data underlying the findings in their manuscript fully available?

Reviewer #2: Yes

5. Is the manuscript presented in an intelligible fashion and written in standard English?

Reviewer #2: Yes

6. Review Comments to the Author

Reviewer #2: I only have a few questions about this revision:

I am not sure if I am allowed to comment on the authors' response to the other reviewer's question, but I saw such statement: "... identifies some interspecific species at genus-level ..." I simply never heard of "interspecific species".

Figure 4A legend: Please add a sentence explaining that genes with zero SNP were not shown.

Line 62-64: The sentence is confusing. I think "X is limited in high-resolution identification" means X markers can only be used to resolve the relationship among species within the same genus, but not the relationship among genus or families? Is this what the authors want to say? If true, is the main purpose of this study to resolve higher level taxonomy? Please clarify this. The fact that I, the other reviewer, and your English editor all are confused by this sentence suggests this needs to be rephrased.

7. PLOS authors have the option to publish the peer review history of their article (what does this mean?). If published, this will include your full peer review and any attached files.

Reviewer #2: No

---

## [Author Response · Author response to Decision Letter 1]

6 Jul 2020

Reviewers' comments:

Reviewer's Responses to Questions

Comments to the Author

1. If the authors have adequately addressed your comments raised in a previous round of review and you feel that this manuscript is now acceptable for publication, you may indicate that here to bypass the “Comments to the Author” section, enter your conflict of interest statement in the “Confidential to Editor” section, and submit your "Accept" recommendation.

Reviewer #2: All comments have been addressed

2. Is the manuscript technically sound, and do the data support the conclusions?

Reviewer #2: Yes

3. Has the statistical analysis been performed appropriately and rigorously? 

Reviewer #2: Yes

4. Have the authors made all data underlying the findings in their manuscript fully available?

Reviewer #2: Yes

5. Is the manuscript presented in an intelligible fashion and written in standard English?

PLOS ONE does not copy edit accepted manuscripts, so the language in submitted articles must be clear, correct, and unambiguous. Any typographical or grammatical errors should be corrected at revision, so please note any specific errors here.

Reviewer #2: Yes

6. Review Comments to the Author

Reviewer #2: I only have a few questions about this revision:

I am not sure if I am allowed to comment on the authors' response to the other reviewer's question, but I saw such statement: "... identifies some interspecific species at genus-level ..." I simply never heard of "interspecific species".

Response: Sorry, we used the wrong words “"interspecific species” in the statement. The correct words should be “identifies some interspecific relationships at genus-level”. 

Figure 4A legend: Please add a sentence explaining that genes with zero SNP were not shown.

Response: Yes, we agree. We add the sentence that the genes with zero SNP were not shown in Figure 4A.

Line 62-64: The sentence is confusing. I think "X is limited in high-resolution identification" means X markers can only be used to resolve the relationship among species within the same genus, but not the relationship among genus or families? Is this what the authors want to say? If true, is the main purpose of this study to resolve higher level taxonomy? Please clarify this. The fact that I, the other reviewer, and your English editor all are confused by this sentence suggests this needs to be rephrased.

Response: We revised the lines 62-64 as follows:

These analyses have succeeded in clarifying the phylogenetic relationships and degrees of variation among Zingiber species, but in general have been limited in breadth of resolution. 

7. PLOS authors have the option to publish the peer review history of their article (what does this mean?). If published, this will include your full peer review and any attached files.

Do you want your identity to be public for this peer review? For information about this choice, including consent withdrawal, please see our Privacy Policy.

Reviewer #2: No

Response: We used PACE to change the figure files to meet PLOS requirements one by one.

---

## [Decision Letter · Decision Letter 2]

10 Jul 2020

Complete chloroplast genomes of Zingiber montanum and Zingiber zerumbet: genome structure, comparative and phylogenetic analyses

PONE-D-20-07989R2

Dear Dr. Li,

We’re pleased to inform you that your manuscript has been judged scientifically suitable for publication and will be formally accepted for publication once it meets all outstanding technical requirements.

Kind regards,

Tzen-Yuh Chiang

Academic Editor

PLOS ONE

Additional Editor Comments (optional):

Reviewers' comments:

Reviewer's Responses to Questions

**Comments to the Author**

1. If the authors have adequately addressed your comments raised in a previous round of review and you feel that this manuscript is now acceptable for publication, you may indicate that here to bypass the “Comments to the Author” section, enter your conflict of interest statement in the “Confidential to Editor” section, and submit your "Accept" recommendation.

Reviewer #2: All comments have been addressed

2. Is the manuscript technically sound, and do the data support the conclusions?

Reviewer #2: Yes

3. Has the statistical analysis been performed appropriately and rigorously? 

Reviewer #2: Yes

4. Have the authors made all data underlying the findings in their manuscript fully available?

Reviewer #2: Yes

5. Is the manuscript presented in an intelligible fashion and written in standard English?

Reviewer #2: Yes

6. Review Comments to the Author

Reviewer #2: I have no other comments. The authors have addressed all my previous comments. Why does the system require a minimum of 100 characters in this part?

7. PLOS authors have the option to publish the peer review history of their article (what does this mean?). If published, this will include your full peer review and any attached files.

Reviewer #2: No

---

## [Editor Report · Acceptance letter]

15 Jul 2020

PONE-D-20-07989R2 

 Complete chloroplast genomes of *Zingiber montanum* and *Zingiber zerumbet*: genome structure, comparative and phylogenetic analyses 

Dear Dr. Li:

I'm pleased to inform you that your manuscript has been deemed suitable for publication in PLOS ONE. Congratulations! Your manuscript is now with our production department. 

Kind regards, 

on behalf of

Dr. Tzen-Yuh Chiang 

Academic Editor

PLOS ONE